# Strengthening primary health care in Ethiopia: A scoping review of successes, challenges, and pathways towards universal health coverage using the WHO monitoring framework

**Tesfaye S. Mengistu**[1,2*], **Aklilu Endalamaw**[1,2], **Anteneh Zewdie**[3], **Eskinder Wolka**[3], **Yibeltal Assefa**[2]

**1** College of Medicine and Health Sciences, Bahir Dar University, Bahir Dar, Ethiopia, **2** School of Public Health, The University of Queensland, Brisbane, Australia, **3** International Institute for Primary Health Care in Ethiopia, Addis Ababa, Ethiopia

* tesfayesetegn@yahoo.com

## Abstract

### Introduction

The ultimate goal of primary health care (PHC), as a whole-of-government and whole-of-society approach, is to achieve the highest level of health by bringing health services closer to the users. This entails that PHC should be viewed as the all-inclusive strategy to achieve universal health coverage (UHC) of the sustainable development goals (SDG). Ethiopia has been implementing PHC since the Alma-Ata Declaration. The World Health Organization (WHO) has recently released a PHC Monitoring Framework to support the monitoring of progress in PHC implementation. However, an evidence gap highlights the need for studies investigating PHC progress towards UHC using this progress monitoring framework. This study aims to evaluate Ethiopia's PHC system using the WHO PHC monitoring framework and identify successes and challenges towards UHC and health security.

### Method

This scoping review was conducted and structured based on Arksey and O'Malley's methodological framework. We searched five databases (PubMed, Scopus, Embase, Web of Science, CINAHL) and hand-searched for relevant articles. We used the WHO PHC monitoring conceptual framework to summarise findings qualitatively. We reported our findings using the Preferred Reporting Items for Systematic Reviews and Meta-Analyses Extension for Scoping Reviews (PRISMA-ScR) framework.

### Results

We included a total of 110 papers – 56 cross-sectional/national surveys, 19 qualitative studies, 16 mixed-method studies, five fiscal/cost/formative or project model analyses, three ecological/ethnographic studies, three longitudinal/quasi-experimental studies, and

**Data availability statement:** All relevant data are within the paper and its Supporting Information files.

**Funding:** The authors received no specific funding for this work.

**Competing interests:** The authors have declared that no competing interests exist.

two each of implementation/participatory research, cohort studies, and case studies. The Ethiopian PHC system has achieved encouraging success in improving healthcare access and coverage, driven by growing political and leadership commitments through implementing a national health extension package (HEP), service integration and multisectoral approaches to achieve UHC. However, Ethiopia's efforts to achieve UHC have faced many challenges, including inadequate service integration, lack of resources and budgets, uneven distribution of health workers and infrastructure, gaps in priority setting, service innovation, stakeholder engagement and funding PHC research. These are affecting access to affordable care and hindering the progress towards UHC.

## Conclusion

Ethiopia's PHC system has achieved significant progress in expanding infrastructure and improving access to health services towards UHC. However, challenges remain, particularly in underserved rural areas, with inequitable access, weak governance, and limited integration of essential services. Hence, by improving resource allocation, addressing rural inequities, systemic and infrastructural challenges and fostering stronger governance and service integration, Ethiopia can further improve and build on the successes of the PHC system, making it more resilient and better equipped to meet the health needs of its population.

## Introduction

Primary health care (PHC) is a whole-of-government and whole-of-society approach to ensure the highest possible level of health and well-being for all individuals and communities and to achieve health equity [1]. Recognising the importance of PHC, Ethiopia adopted the PHC system in 1978 as a national strategy to make health services more accessible and increase service coverage through community participation and multisectoral actions. The adoption and progress of PHC in Ethiopia has socio-political, economic and developmental underpinnings [2,3]. Since its adoption, the Ethiopian PHC system approach has shifted from an urban-centric, curative-oriented focus to a whole-of-society approach, including rural and remote populations with a disease prevention orientation that has remained resource-constrained [4].

Since the 1990s, major health policies and strategic initiatives have been implemented [5,6]. These efforts focused on expanding healthcare infrastructure, improving healthcare access, training the workforce, and introducing a community-based health insurance scheme. As a result, service coverage and quality of care improved, leading to better health outcomes and reduced disparities among the diverse population [2,7,8]. The government's commitment and effective implementation of the Health Sector Transformation Plans (HSTP) is vital to strengthening Ethiopia's PHC system to improve access and coverage and provide equitable and quality services to achieve universal health coverage (UHC) [5,9–11]. In addition, the national government enforces community ownership and accountability and puts forward strategic partnerships with multi-lateral international donors, development organisations and non-governmental organisations, which are other important factors for the success of PHC [9,12,13].

The national focus on addressing critical system-related and operational challenges of PHC service through multisectoral collaboration has contributed to the rapid expansion of PHC [14–16] towards achieving UHC in Ethiopia. This strategic focus on multisectoral PHC policy and action helps to optimise the gains from PHC. These strategic commitments facilitated

multi-lateral governmental and non-governmental organisations to provide financial and technical support, capacity development and knowledge exchange support to the Ethiopian healthcare reforms. The national focus on critical challenges on PHC and multi-sectoral collaboration greatly supported the reforms by increasing financial and technical resource flows, capacity development and knowledge exchange, accelerating the progress towards UHC.

The rapid expansion of PHC in Ethiopia over the past 15 years is recognised as a model for the rest of the Sub-Sahara countries [14]. Ethiopia's trajectory to UHC has been underpinned by strong political commitment, community engagement, strategic partnerships and milestones in rolling out PHC system [17]. This commitment to expanding PHC by developing a pioneering Health Extension Program (HEP) has put Ethiopia among the nations with considerable steps towards UHC [12,18].

Previous studies have investigated the successes and challenges of PHC [5,9,13] and achievements of the PHC system, including health service coverage [19–23], quality of care [24–29], PHC programmes/strategies [9,12] and its capacity to achieve UHC [10,30]. Although they provided useful information, many previous monitoring and evaluation frameworks lack uniformity, show data discrepancies, and use inadequate indicators [31]. This highlights the need for comprehensive evidence based on a structured framework to provide holistic progress of PHC systems towards UHC in Ethiopia is critically needed. This scoping review aims to synthesise the available evidence and identify the successes and challenges of PHC system efforts towards UHC in Ethiopia using the WHO's monitoring framework.

## Methods

This is a scoping review of the success and challenges of Ethiopian PHC in achieving UHC developed and structured based on the Arksey and O'Malley framework [32]. Arksey and O'Malley recommend using six (five compulsory and one optional) stages of scoping review to maximise the usefulness and rigour of study findings [33]. The compulsory stages include identifying the research question, identifying relevant studies, selecting studies, charting data, collating, summarising, and reporting results, and consultation (optional) (S3 Table) [34,35]. Each compulsory stage is outlined below.

### Stage 1: Identifying the research question

Our research question for this review is: What are PHC's successes and challenges toward achieving UHC and health security in Ethiopia?

### Stage 2: Identifying relevant studies

We searched five databases (PubMed, Scopus, Embase, Web of Science, CINAHL) from June 6 to June 13, 2024, without restricting the publication period. We developed our search strategy based on the "primary health care measurement framework and indicators" [36]. We included search terms: integrated health service*, multisectoral polic*, multi sectoral polic*, multi-sectoral action*, multi sectoral actio*, primary health care, primary healthcare, primary care, community involvement, health, Ethiopia*. The search terms were combined by the Boolean operators "AND" or "OR", and the search strategies were adapted to the specific electronic databases by modifying field codes. TSM conducted database searches. The search terms, search strategies for each electronic database and filters are presented in the S1 Table.

### Stage 3: Selection of studies

We imported all retrieved citations into EndNote X20 and deleted duplicates. Two authors performed screening independently. The first author (TSM) performed the title and abstract

screening and synthesised a final list of the papers. Then, AE and the senior author (YA) screened the final list of papers deemed eligible for full-text review. We resolved disagreements through discussion.

We included only studies published in English and conducted in Ethiopia that met the following criteria: (1) Studies of any design without year of publication restriction and (2) studies conducted on the subject area of PHC regardless of the health care professional categories involved. We excluded methodology papers/protocols, global studies, conference/workshops/ seminar papers, editorials, systematic reviews, protocol/pilot studies/tool and validation studies.

### Stage 4: Charting the data

We developed a data-charting form covering study characteristics (author, year of publication, study type/design, study participants, key concepts, and main findings from each study (S2 Table). We also extracted the successes and challenges of PHC in Ethiopia using the WHO PHC performance conceptual framework. TSM extracted data using a data–charting sheet. Two authors (AE and YA) checked the extracted data, content completeness, accuracy, and quality.

### Stage 5: Collating, summarising, and reporting results

We summarised the research findings using the data extraction sheet. Data were synthesised thematically and narrated using the key focus (themes) of the PHC monitoring framework and PHC strategic and operational levers (Fig 1). The successes and challenges/barriers to PHC service are presented. This scoping review was conducted and reported according to the Preferred Reporting Items for Systematic Reviews and Meta-Analyses Extension for Scoping Reviews (PRISMA-ScR) (S3 Table) [37].

### Conceptual framework

We used the WHO PHC monitoring conceptual framework [36] to assess the success and challenges of PHC systems in Ethiopia. This framework is organised to support the PHC theory of change, providing a logical, results-based structure for monitoring performance and progress in PHC towards UHC goals. It illustrates the logical relationships between domains, demonstrating a causal pathway linking PHC structures, inputs, and processes to desired results. We used strategic and operational levers to describe the successes and challenges of the Ethiopian PHC system and how the system contributes to UHC (Fig 1).

## Results

In total, we retrieved 3,657 records from five electronic databases. After removing duplicates (n = 1,501) and studies with non-relevant titles (n = 2,156), 171 eligible studies were reviewed in full-text screening. After full-text screening, 68 studies were excluded, for 103 articles in total. We further identified seven additional research articles via hand-searching. Altogether, 110 studies were included in the review. The detailed screening, study selection process and reasons for exclusion are shown in Fig 2.

### Characteristics of included studies

The characteristics and key findings of the included studies are summarised in S2 Table. Of the total 110 included studies, 50.1% (n=56) were cross-sectional/national survey studies [8,16,19,21–23,25–29,38–83], 17.3% (n = 19) were qualitative studies [5,6,10,15,17,84–97]

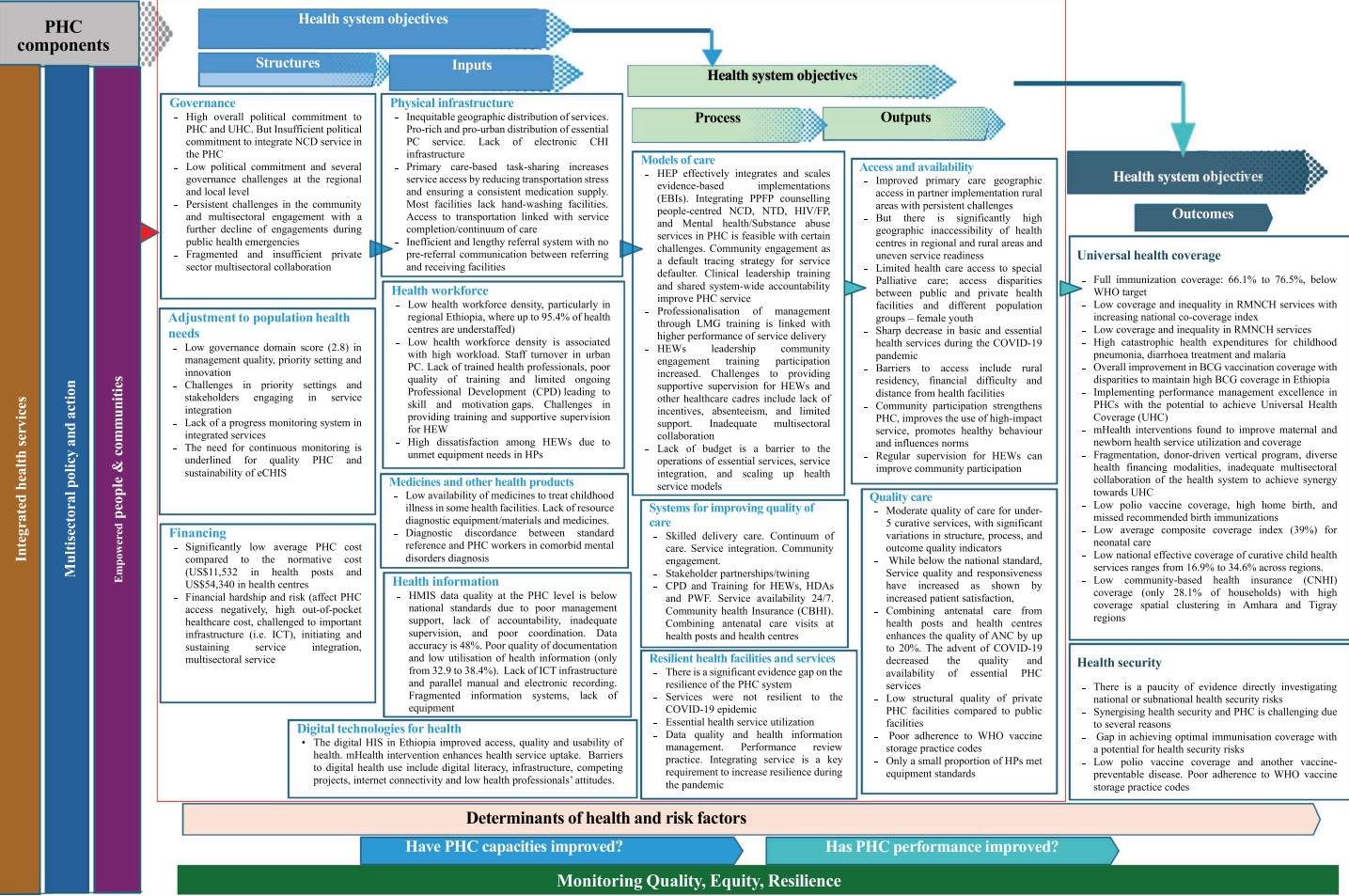

**Fig 1. WHOs primary health performance monitoring framework.**

including document reviews and critical interpretive analysis, 14.5% (n = 16) were mixed studies [9,11,30,98–109], five were fiscal/cost/formative or project model analysis [110–114], three were ecological/ethnographic studies [20,115,116], three were longitudinal/quasi-experimental studies [117–119], and two each implementation/participatory research [120,121], cohort studies [122,123], and case studies [13,124].

## Successes of PHC in Ethiopia

Table 1 summarises the observed successes of PHC towards UHC in this scoping review. Our study shows that there is significant progress in advancing PHC towards UHC. There was increased geographic access to PHC facilities with associated health service utilisation, coverage, and improved quality of care. The explanations for the observed success include improved political and leadership commitment with improved governance, increasing emphasis on multisectoral action and health service integration, and ever-increasing mobilisation for community engagement (Table 1).

## Findings based on the WHO PHC monitoring framework

In Fig 1, we summarised our detailed findings on the Ethiopian PHC system structured based on the WHO PHC Monitoring Framework. The figure provides a clear snapshot of the

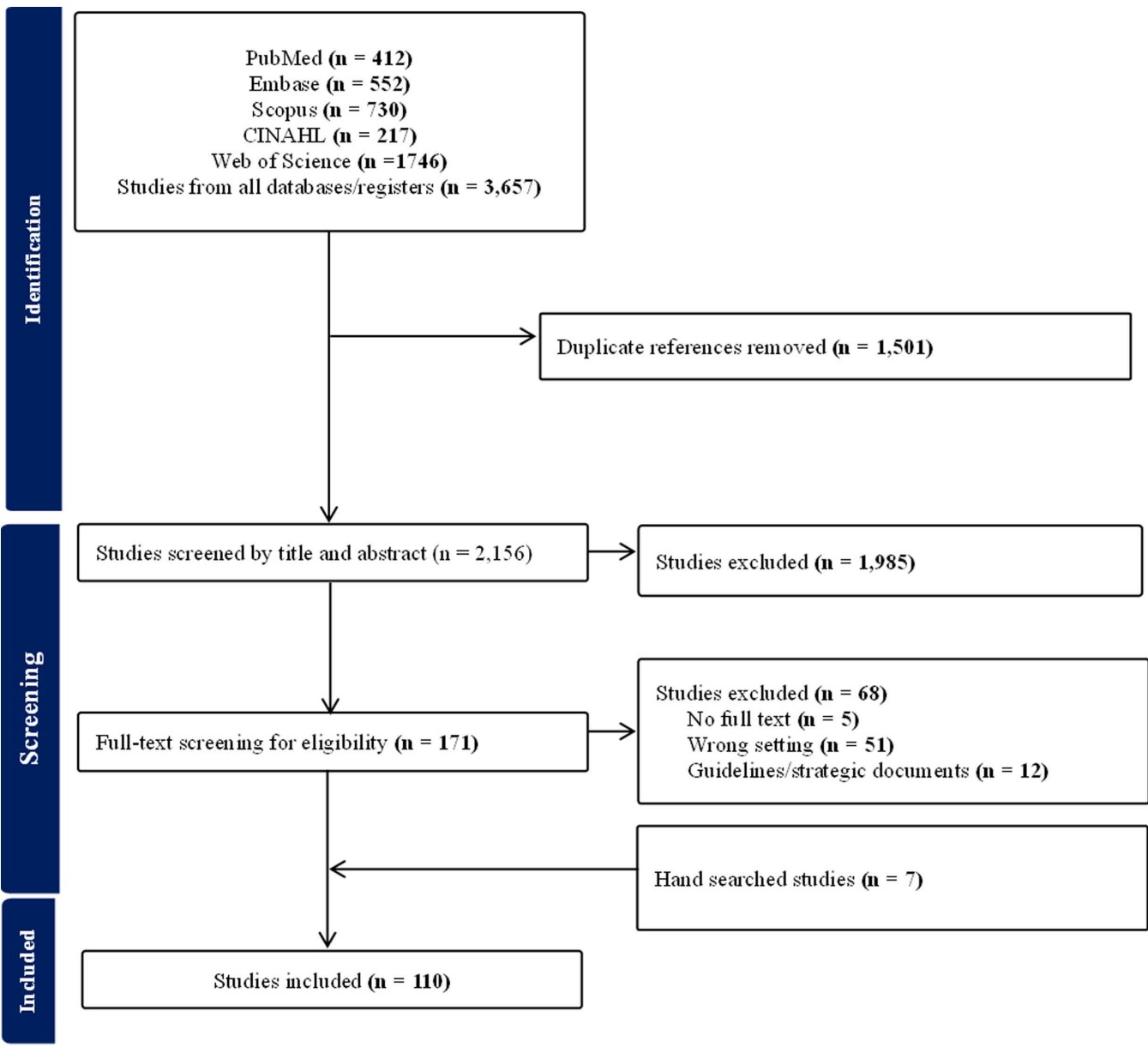

**Fig 2.  PRISMA flow chart.**

progress and ongoing challenges of the Ethiopian PHC system towards achieving UHC and health security.

## Structure of the PHC system

**Governance of PHC.** In this scoping review, eight studies [5,30,81,92,96,117,125,126] reported on PHC governance. In Ethiopia, there is significant political and leadership commitment to implement PHC towards attaining UHC [5,30,92,125]. For example, the governance of Ethiopia showed significant and transferable achievements in initiating family planning [118], sexual and reproductive health [17], HIV services [61] and mental health services [92,96] integration into the PHC system. The National Health Extension Package (HEP)

**Table 1. The success of the PHC system in Ethiopia.**

| Success parameters | Success observed in the PHC system |
|---|---|
| Governance and political commitment | • Political commitment and favourable macro-fiscal environment for health investments<br>• Progress in policy development, engagement in service delivery in private practices<br>• Improved quality in management, inputs, and population health management<br>• Significant progress in NCD policy development and activities to reduce risk factors<br>• Significant improvement in organisational culture and excellence scores<br>• Significant improvement in adherence to management standards and healthcare performance |
| Access to PHC facilities | • Improved geographic access to health facilities in rural areas<br>• Increased access, reduced cost, and reduced transportation stress |
| PHC service coverage | • Increasing immunisation service and coverage<br>• Significant improvement in BCG immunisation<br>• High PCV-10 coverage (three doses) (76%) with high acceptability (nearly 100%) |
| Health workforce training and CPD | • Continued commitment to increase health human force training programs<br>• Significant improvement in community health worker training attendance, enhancing community engagement.<br>• Leadership, management, and governance training improved district health and health system performance<br>• Continued government commitment training and empowering health extension workers and local health workers |
| Healthcare utilisation | - Increased utilisation of antenatal care and skilled delivery services<br>- mHealth intervention significantly increased maternal and newborn health service utilisation<br>- Mothers Waiting for Home improved immediate uptake of postpartum family planning<br>- Effective counselling in antenatal and postnatal care increases PPFP uptake |
| Community participation/ engagement in PHC | • Improved community and government participation in strengthening primary healthcare<br>• Community engagement improves service use and coverage as it facilitates defaulter-tracing<br>• Improved community understanding of mental health |
| PHC service integration | • Improvements in service integration of health strategies<br>• Progress in integrating SRHR services into UHC<br>• Successful integration of MH/SA services with improved quality and outcomes<br>• Integration/ Combining care settings enhances quality<br>• Integration increases the likelihood of clients initiating HIV testing and attracts diverse client types<br>• Improvements in service integration with impacts on coverage<br>• Integration is feasible, acceptable, and cost-effective |
| Quality of care | • Satisfaction with primary healthcare services is generally high<br>• Improvements observed, but challenges remain.<br>• Moderate quality with significant variations in structure, process, and outcome components and significant regional variations |

played a significant role in initiating and leveraging the success towards service integration [121]. However, there is still inadequate commitment and lack of prioritising to integrate many essential health services, including non-communicable disease (NCD) prevention and management service [10] and palliative care [84] service in PHC and allocates inadequate resources [125,126]. Furthermore, measuring the PHC achievements towards UHC remains challenging due to the lack of a progress monitoring system [81]. There are also challenges related to governance and political commitment at lower governance and leadership structures, including at the sub-national, regional and district levels [30,117]. Ensuring community and multisectoral engagement in PHC governance also faces persistent challenges [6,112,115]. Community engagement and public-private sector collaboration remain inconsistent and fragmented [69,96], which further declined during the COVID-19 pandemic [93].

**Adjustment to population health needs.** Six studies [6,28,30,73,81] described the importance of tailoring healthcare services and interventions to meet the specific health needs of a population. Although there is limited evidence, there is a low governance domain score in quality management [30] to meet different communities' unique health challenges and priorities. Priority setting [30,81], service innovation [30] and engaging stakeholders in service expansion and integration [81] remain the challenges of the PHC system in Ethiopia. There are also challenges in implementing continuous monitoring and evaluation to ensure quality

PHC [28] and sustainability of integrated services [6]. However, there is an evidence gap in the amount of funds allocated to PHC research.

**Primary health care financing.** Nine studies [13,46,75,77,78,80,107,114,115] described financing and cost-related components of PHC. The average PHC cost was US$11,532 in health posts and US$54,340 in health centers. This figure is significantly low compared to the normative average cost [114]. Financial hardship in accessing PHC [78], high out-of-pocket healthcare costs [75,77], and financial challenges to render important infrastructure(i.e., ICT) [115] are still daunting challenges in the Ethiopian PHC system. Particularly, the implementation and sustainability of service integration [107] and rendering multisectoral service highly depend on effective financial governance, which requires political commitment [13,77]. The PHC system in regional and remote areas has financial challenges in introducing essential services and providing collaborative and multisectoral services [80].

## Inputs of the PHC system

**Physical infrastructure.** Six studies [21,51,54,63,92,96] described the distribution and density of PHC physical infrastructures in Ethiopia. While there has been substantial progress in the development of physical infrastructures, this review shows a significantly high inequitable distribution of healthcare infrastructure [63,127]. The density and distribution of physical infrastructures and essential PCs in Ethiopia tend to be pro-rich and pro-urban [21], critical low investment in physical infrastructure [128] further exacerbating inequities in healthcare infrastructure availability and health service access. Furthermore, most available PHC facilities lack adequate Water, Sanitation, and Hygiene (WASH) facilities [28].

**PHC health workforce.** Sixteen studies [6,10,13,16,27,28,54,73,81,84,88,91,102,103,105,122] described health workforce density, distribution and professional development issues in the PHC system. This scoping review shows that the national health workforce is unevenly distributed, with significant disparities in staffing across regions [27,73]. For example, 95.4% of health centers in the Somali region are understaffed, with a critical shortage of pharmacists and laboratory technicians [73]. Although the health workforce is overly populated in urban health institutions, high workload, staff burnout, and turnover remain a key barrier to primary care service [6,84,122]. While it is projected to increase across all areas due to population growth and changing disease patterns, the health workforce workload varied significantly between regions [16].

Lack of a trained PHC workforce [10,81,88] combined with poor quality of available training [91] and limited ongoing Professional Development (CPD) for health workers [28] are critical challenges directly affecting the quality of care and accessibility of health services nationwide. Our review also emphasised the low satisfaction score, essential gaps in skill and motivation of HEWs [102] combined with challenges in providing training and supportive supervision for HEWs [105] that potentially impede the progress of achieving UHC.

**Medicines and other health products.** Seven studies [10,15,54,67,84,88,124] reported the availability, utilisation and challenges related to medicines, diagnostic equipment and other essential resources of the primary health care (PHC) system. In our review, a fiscal space analysis shows the largest share, 50–70% of health costs, were allocated for medicines, commodities, and supplies [113]. Contrary to this, other studies also reported that only a small proportion of PHC settings met equipment standards [102], lacked essential health resources [54,124], diagnostic equipment/materials [15,84,88] and medicines [10,67] associated with significantly higher dissatisfaction among HEWs.

**Health information.** Nine studies [10,12,25,45,46,88,91,95,99] reported findings related to health information infrastructure, information management systems, data quality and utilisation. In our review, PHC facilities are characterised by low HMIS data quality [25], low

data accuracy [25], low health information utilisation (only from 32.9 to 38.4%) [45,46,99], poor quality of documentation [95] and fragmented reporting [10,25,99]. This is due, in part, to the lack of ICT infrastructure/equipment [6,10,99] and parallel manual and electronic recording [91], poor management support, lack of accountability, inadequate supervision and poor coordination [25]. Perceived high workload [6,46], data management knowledge [46], staff turnover [6], access to HMIS resources [46] and lack of ownership [6] affect health information use and implementation of health information and management initiatives in PHC.

**Digital technologies for health.** Five studies [6,129–132] described digital literacy, digital infrastructure, the use of digital health, and barriers to using digital health in PHC. Our review shows that, despite limited studies, mHealth intervention enhances health service uptake, such as antenatal and neonatal health service uptake. Particularly during epidemics (e.g., COVID-19), using digital technologies for health could make the health system more resilient [133,134]. However, users' digital literacy, critical digital infrastructure problems, competing projects, lack of wide range internet connectivity and low health professionals' attitude [130] are major barriers to digital health use in Ethiopia [6,131,132].

## PHC service delivery process

**Models of care.** Eleven studies [58,81,84,88,90,99,105,110,112,118,121] described different models of care service improvement parameters, including evidence-based care [121], integrated service models [58,84,88,99,118] people-centred services [81,88], community engagement for service provision [112], leadership improvements for service provision [81,105] and system-wide accountability to improve service [90]. This review highlighted that HEP, as a service provision model, is a feasible entry point to integrate many primary care services, including PPFP counselling [118], people-centred NCD [10,81,88], NTD [88,99], HIV/FP [58] and mental health/substance abuse services [92,125]. Community engagement at the PHC units is a promising strategy to improve the performance of the primary care system. For example, we observed that community engagement facilitates health service defaulter-tracing [112].

Our study also shows that continuous leadership, management, and governance training for PHC staff increases their management professionalisation for higher service performance and community engagement [11,53,112]. In our review, we observed that HEW leadership community engagement training and participation increased [105] with training in clinical leadership [81] and shared system-wide accountability found to improve PHC service [90].

A few studies also show that PHC units can effectively integrate services and leverage evidence-based implementations (EBIs) [121] nationally. However, providing supportive supervision for HEWs [92,105] and other healthcare cadres [56] remains challenging due to a lack of incentives, absenteeism, and limited support. Additionally, inadequate multisectoral collaboration [13] and lack of budget are barriers to provide essential services [113], implement service integration [88], and scaling up health service models [110]. For some of the services, while the preferred integration modality (co-locating vs. same rooms) needs more research, a study reported that same-room integration of services, e.g., HIV and FP, is 2–13 times more likely to increase service use [58]. Ensuring stakeholder partnerships/twining [9,112] and acceptance, gradual integration, supervision/mentorship and ongoing training [28,92,96,112], leveraging resources from SDG programs [125] are crucial for sustainable care delivery to avoid "task dumping" [92] and accelerate UHC in Ethiopia [9].

This review also demonstrated that the referral system in PHC is inefficient [116] and lengthy [72], with a lack of pre-referral communication between referring and receiving facilities [54,116]. However, a study reported that task-sharing between primary care settings and

hospitals preceded by adequate training and support is feasible to increase service access by reducing transportation stress and costs [120] as it avoids referral for basic and essential PHC services. This study also shows that task-sharing ensures a consistent medication supply [120]. The increased accessibility of PC service due to task-sharing ensures a continuum of care [51], supports service changes and facilitates community resource mobilisation [92,120]. A well-designed, effectively planned primary care-based task-sharing model that leverages available resources is a critical success factor for service integration [96,135] and could accelerate the achievement of UHC [136].

**Systems for improving quality of care.**  In this study, we identified skilled delivery care [49], service integration [21,58,81,85,86,96], community engagement [112], stakeholder partnerships/twining [9], training including continuous professional development (CPD) [28], Leadership Governance and Management (LMG) [53] and training for HEWs [92,96], Health Development Armies (HDAs) and Pregnant Women's Forum (PWF) [105], 24/7 service availability [29], community-based health insurance (CBHI) [8] and combining antenatal care visits at health posts and health centers [66] were systems established to improve the quality of primary care.

The link between skilled birth care and quality indicators of maternal health service is further enhanced with community participation in skilled birth care [49]. Health workers LMG [53] training and training HEWs [92,96], HDAs and PWF [105] is associated with significantly higher ANC follow-up retention [53,105] and institutional delivery [53] in primary care [53,105]. However, other studies show an ineffective continuum of care characterised by poor documentation quality and inadequate advocacy [95]. Our study identified service integration [21,58,81,85,86,96] is promising to achieve UHC; however, integrating special care such as palliative care is not well integrated into the health care system mainly due to lack of priority, low health professionals and budgets [86].

**Resilient health facilities and services.**  In this review, three studies assessed the resilience of primary care services during public health emergencies - COVID-19 pandemic [55,101,123] that primary care services, including essential health services, quality of data, information management and performance review practices, were not resilient to the COVID-19 epidemic. For example, while some essential services are unaffected or recovered quickly, sharp inpatient and outpatient decline was observed [123]. This underscores the urgent need to assess health system resilience further along the entire PHC spectrum. A study also suggested that well-integrated service could increase health system resilience [101]. This review shows a critical evidence gap regarding the resilience of the PHC system and primary care units, including the percentage of facilities that meet the criteria for a resilient health facility or service.

**Access and availability.**  Our review found an overall improvement in access to PHC services that could contribute to UHC, likely due to the presence of PHC policy and leadership structures at the federal and state level [30]. However, our study shows that PHC service accessibility remained very low in regional and rural areas [27,73,78] with significant inequality in service readiness [56,70] and accessibility due to persistent challenges/barriers affecting primary healthcare service access [5,27,28,73]. We also observed significant access differences between public and private health facilities [29] and population groups where youth services are less accessible to female youths [78]. For example, up to 65% of the population in the Somali region had no access to PHC centers with the required health workforce [73].

Given that the distribution of PHC facilities is pro-rich and pro-urban [21], the availability and accessibility of health-centre-based PHC and skilled health workforce is characterised by access inequality [27]. Our study also shows limited healthcare access to special care (palliative care [84]) due to a lack of priority. Our study also shows that primary care access is

significantly influenced by the COVID-19 pandemic [101,123]. This review further revealed that increased community participation improves access to PHC services [126], increases the use of high-impact services [49], and ensures healthy behaviours and norms [126], enabled by regular supervision for HEWs [105].

**Quality of care.** In this review, we observed that PHC service quality and responsiveness show improvements [28,64]. Our study also shows good/moderate quality of maternal [64] and child health service [23] to high patient satisfaction even during COVID-19 [28]. However, other studies show that the quality of mental health services during the advent of COVID-19 decreased significantly due to task and resource shifting [101]. Service quality is also a primary reason for the low or non-utilisation of essential services for child health [48]. Patient satisfaction as a measure of quality of care, for example, for IMNCI, is below the national standard due to waiting time and availability of medications [67]. There are significant variations in structure, process, and outcome quality indicators [23] across PHC facilities. The difference in structural quality components is worse in private PHC facilities compared to public facilities [29].

We identified multiple barriers affecting the quality of care and implemented initiatives to ensure the quality of care in PHC units, including low government budget, high user cost [88,113,114], less attention [88,125], lack of trained professionals [29,88], resource gaps and unmet resource standards [69,102,113]. Studies have also reported poor adherence to WHO vaccine storage practice codes [69] and unmet equipment standards in the majority of HPs [102]. In 2023, a study conducted in nine regions reported that the overall resource gap is significantly lower compared to the HSTP II targets for 2024/25, with observed resource gaps of 53–75% in health posts and 39–83% in health centers [114]. The implementation of an essential health service package (EHSP) to ensure quality care is significantly affected by a 33% resource gap against SDG [113]. This review also highlighted that integration of services was highly associated with better quality of care and health system performance. Integration of antenatal care services between health posts and health centers improved the quality of antenatal care (ANC) by 20% [66]. Likewise, twinning partnerships have been shown to improve district health system performance towards UHC [9].

## Outcomes of the PHC system

**Universal health coverage.** Our review shows that the overall coverage of PHC services is still low [19,21,22], with a low service coverage index [62]. This review revealed a gap in achieving optimal immunisation coverage, which ranges from 66.1% to 76.5%, below the WHO target [19,22] and significant urban vs. rural inequality [19]. Our synthesis also shows low polio vaccine coverage due to high rates of home births and missed recommended immunisations [39] alongside a 47.6% increase in national BCG vaccination coverage [20]. Despite a significant overall improvement in BCG vaccination coverage in Ethiopia, persistent geographical disparities highlight the need for targeted interventions to maintain high coverage and prevent tuberculosis effectively [20].

There is low national coverage and inequality in RMNCH services [21,26,62], quality intrapartum care [26], and curative child health services [23] and the highest inequality observed for different services and across regions [23] though there is a significant increase in the national co-coverage index (2005–2019) [21]. The national coverage of community-based health insurance (CBHI) still sits at 28.1%, with high coverage spatial clustering in the Amhara and Tigray regions [63], leading to high catastrophic health expenditures for basic and essential childhood curative PHC service [75,77]. In addition to challenges related to inequality [80], and resource limitations to improve national PHC service coverage, fragmentation, donor-driven vertical program, diverse health financing modalities, and inadequate

multisectoral collaboration of the health system further affect the achievement of PHC services synergy towards UHC [13]. However, using mHealth interventions and implementing performance management in PHCs have a high potential to enhance UHC [74].

**Health security.** There is a substantial evidence gap related to health security in PHC. The limited evidence shows that integration of health security measures with PHC is fraught with challenges due to several factors [13]. This could be due to a lack of comprehensive data on health security threats and risk behaviours, which makes it difficult to develop targeted interventions. A study indicated that synergising health security with PHC is hindered by fragmented health systems, limited resources, and inadequate multisectoral collaboration [13]. In this review, we speculated that low polio vaccine [39] and full immunisation coverage compared to the WHO target [19,22], and significant geographical inequality in the coverage of vaccine-preventable disease [20] coupled with poor adherence to WHO vaccine storage practice codes [69] are potential risks for future health security. The unmet resource needs in the majority of HPs [102] make timely detection of health security risks challenging.

## Discussion

In this review, we explored the key successes and challenges of the Ethiopian PHC in achieving UHC and health security using the WHO's PHC performance monitoring framework. To the best of our knowledge, this is the first to comprehensively review Ethiopia's PHC systems and identify successes and challenges using the PHC strategic and operational levers monitoring indicators. The WHO PHC monitoring conceptual framework indicated that integrated health service is one of the PHC components designed to achieve UHC. This framework also recommends an effective integration between strategic levers and operational levers, considering primary care and essential public health functions [36].

The Ethiopian PHC governance decentralised system ensures equitable service delivery through the Primary Health Care Units (PHCUs) - comprising five satellite health posts, referral health centres, and primary hospitals. In accordance with the WHO PHC monitoring framework, this scoping review shows there is a clear government and leadership commitment to adopting and implementing local and international PHC strategic initiatives [30,125] by prioritising essential health service expansion and integration to achieve UHC [5,30,92,125]. Service integration increases access and coverage to meet the changing health needs of the population in the context of dynamic socio-economic and political contexts and increasing disease burden and co-morbid conditions [137,138]. Cognizant of this, the government of Ethiopia has given priority to strengthening the PHC systems and integrating essential health services to achieve UHC [5]. Yet, national evidence [19,21,22] and regional UHC progress monitoring data [139] shows that the overall PHC service coverage is still low, with a low service coverage index [62], making Ethiopia among the 12 African countries with low coverage (index between 20 and 39) [139]. Similarly, a recent Woreda-level analysis reveals a service coverage index of 57%, highlighting significant gaps in achieving UHC [140].

The Ethiopian PHC system has achieved significant and transferable achievements in initiating family planning [118], sexual and reproductive health [17], HIV services [61] and mental health [92,96] service integration into PHC. Studies also show that the innovative national HEP played a significant role in initiating and leveraging the PHC service integration [121]. In line with the government's strategic goals to achieve UHC by further increasing service accessibility, we observed several primary care integration activities using HEP as an entry point. In addition to the success achieved so far, there are promising initiatives to integrate many services. These include post-partum family planning (PPFP) counselling [118], people-centred care for non-communicable diseases (NCDs) [10,81,88] and the integration of

neglected tropical diseases (NTDs) [88,99] into PHC, demonstrating significant community and stakeholder engagement. Studies suggest that continuous monitoring and evaluation, prioritising healthcare needs [141,142] and tailoring healthcare services and interventions [30,81] are critical steps in the process of designing and implementing PHC service integration. It is vital that health service users' health needs – the demand and the government's service identification and priority-setting exercise – and the supply of services- be well-aligned for effective service integration and operation [96,135].

Although there is limited evidence, we observed that there is a low governance domain score in quality management [30] to meet the unique health challenges and priorities of different communities. However, our study indicates that there is still inadequate political commitment and a lack of prioritisation in integrating many essential health services [10,84] into primary care and allocating sufficient resources [125,126]. Parallel to the observed political commitment to initiate and sustain service integration, service integration progress monitoring and quality management procedures are equally important. However, in our review, studies show that lack of progress monitoring system [81], low governance score in quality management [28,30], problems with priority setting [30,81], stakeholders' engagement in service expansion and integration [81] and service innovation [30] are persistent challenges of the PHC system in Ethiopia.

Ensuring post-integration PHC service sustainability [6] is a challenging task in resource-limited countries like Ethiopia and needs a parallel commitment from sub-national and regional political leaders and government officers. In our review, some studies identified low governance and political commitment at the regional and district levels [30,117]. Given that these hierarchies are the middle-level and operation levels, low commitment may affect the community and multisectoral engagements in service integration. Yet, ensuring the full engagement of community and multisectoral stakeholders in PHC governance remains a challenge to achieving UHC [6,112,115]. Persistent high inequalities in healthcare infrastructure also affect PHC service integration [63,127,143]. The federal structure in Ethiopia presents both opportunities and challenges to the implementation of PHC towards achieving UHC. For example, in remote and underserved regions such as Somalia [73]. and Afar, regional leadership face financial/budget [80], coordination, infrastructure, staffing, and resource challenges to effectively engage in the implementation of PHC policies. Furthermore, regional and district leaders need to be more actively engaged in advocacy and establish stronger partnerships with the national government for more resources and support. Capacity building of regional and district health leadership through regular training and continuous professional development is also critical to addressing the governance and political commitment gaps [30,117], to enhance the effective implementation of the PHC towards achieving UHC.

Operational-level evidence-based primary care (EBPC) task-sharing can be a key driver for successful PHC service integration [96,135]. Studies in this review also show that adequate stakeholder partnerships/twining [9,112], gradual integration, mentorship and ongoing training [28,92,96,112], and leveraging resources from SDG programs [125] are crucial for sustainable integrated PHC [92] and accelerate UHC in Ethiopia [9]. In parallel with the growing effort to reorient more services into the PHC system, revisiting the dynamic health needs and designing more service quality improvement systems are needed. In this case, reinforcing and marshalling the local and international resources towards community engagement [112], stakeholder partnerships/twining [9], and training including CPD [28], LMG [53] and other training for HEWs [92,96] and other health cadres are highly required. On the other hand, generating evidence to guide policymakers in developing high-impact PHC system integration is critical. Although some studies suggest that single-room service integration is associated with 2–13 times increased service use [58], the preferred integration modality (co-locating vs.

same rooms) is still debatable and worth investigating further. The impact of service integration on the resilience of the PHC system during emerging and re-emerging health security threats hasn't been investigated in Ethiopia. It is also worth investigating how service integration could work best during war and conflict times.

Multisectoral policy and action is the second PHC component suggested by the WHO's PHC monitoring conceptual framework to achieve UHC [36]. Recognising the fact that primary care strengthening requires synergic policies, Ethiopia has a high national level interest in a multisectoral approach to PHC, which has resulted in exemplary success in its PHC system compared to other countries in the continent [13]. The national focus on addressing critical system-related and operational challenges of the PHC system through multisectoral collaboration has contributed to the rapid expansion of PHC towards achieving UHC in Ethiopia [14–16]. The recently developed multisectoral National Action Plan for Health Security (NAPHS) [144] can strengthen health systems and accelerate the UHC. It is also believed that the multisectoral action to develop this policy would increase resource and PHC system resilience in future public health emergencies [144]. Political commitment to financial governance or resource mobilisation is crucial for effective multisectoral action [13,77]. For example, in line with the multisectoral food and nutrition policy, the Government of Ethiopia has prioritised nutrition as a political agenda and allocated significantly high investment in nutrition interventions with its partners [145].

The strategic focus on multisectoral PHC policy and action to optimise the gains from PHC has encouraged multi-lateral governmental and non-governmental organisations to provide financial and technical support, capacity development and knowledge exchange to the Ethiopia healthcare reforms. This shows that quality and affordable PHC service entails multisectoral policy and action. However, this study illustrates that multisectoral engagement [107] and public-private partnerships [69,96] within PHC systems remain fragmented and inconsistent [69,96], which further declined during the COVID-19 pandemic [93]. A multisectoral NAPHS has been recently developed to improve the health system resilience and response based on a lesson learned from COVID-19 response [144]. The multi-sectoral collaborative intervention to end food security and childhood malnutrition by 2030 under the "Seqota Declaration" within the Health Sector Transformation Plan II (HSTP II) is a good example of a multisectoral strategy that would significantly impact the country's health outcomes if implemented at a larger scale [146]. Nevertheless, our study highlighted the evidence gap in multisectoral collaboration and community empowerment, underscoring the need for more studies that explore the role of multisectoral collaboration and community empowerment in achieving UHC using the WHO monitoring framework.

Studies show that meeting the dynamic population health needs throughout the changing socio-economic and political contexts, with a multisectoral approach, is the primary strategic focus of the health system [137,138]. Despite advancements in health outcomes and improvements in social determinants of health, many systems still operate separately. Multisectoral synergy is critical for successfully implementing healthcare reforms and high-impact strategic initiatives, including PHC, to achieve UHC [5,9,13,147]. However, this study highlighted inadequate multisectoral action and collaboration and a lack of budget and financing modalities, which are significant barriers to the effective operation of essential PHC functions [115] and multisectoral collaboration [80]. Improved multisectoral action could improve resource availability, making health services affordable and improving quality. This, in turn, reduces the financial burden of accessing PHC services and out-of-pocket healthcare costs. Contrary to the decreasing incidence of catastrophic health spending across many countries in the WHO African Region [139], we observed high financial hardship in accessing PHC [78] and high out-of-pocket healthcare costs in Ethiopia [75,77,140], which might be explained by the

lack of full government commitment to ensure multisectoral involvement at the policy level and the fragmentation and inconsistency of existing partnerships, in addition to the shortage of policymakers and experts [145]. These factors also impede the scaling up of health service models that require synergy among multiple sectors to achieve UHC. Therefore, functional multisectoral policy and action and training policy experts are urgently needed to achieve UHC targets [107,145].

Despite Ethiopia being a model to other African countries in its PHC implementation success, regional conflicts, large-scale disease outbreaks/epidemics and crises [14], high burden of food insecurity, problems with WASH [28], inequitable physical infrastructure [63,127], problems with digital health and ICT infrastructure [115], less attention to gender equity [148] can show the lack of sustained political commitment and effective financial governance [13,77] to ensure multisectoral approaches to PHC. Service innovation [30] to accommodate multisectoral approaches and engaging stakeholders [81] is a significant challenge, possibly hindering UHC's achievements.

Multisectoral policy and action integrate various stakeholders and processes across macro, meso, and micro levels to provide strategic policy directions and facilitate or support timely and effective decision-making [149]. For example, reliable and timely health information generated at PHC facilities due to multisectoral action on ICT infrastructure and health information facilitate policy-relevant decision-making across the PHC system [150,151] and inform public health risk monitoring [151]. However, there is a huge gap in planning and executing collaborative action to solve major ICT and digital health infrastructural [6,10,99] and systemic challenges, including poor data quality data/accuracy [25,95], poor management support, lack of accountability, inadequate supervision and poor coordination [25], all of which hinder the sustainability of PHC. This could be explained by challenges related to the PHC governance and leadership in enforcing multisectoral actions.

Overall, the Ethiopian PHC could achieve even better results and significantly contribute toward attaining UHC if the government scales up multisectoral coordination for health policies and actions across sectors and strengthens 'political commitment and financial governance' to support PHC. It is also critical to strengthen public-private partnerships, improve training and support for policy experts, and tackle 'systemic and infrastructural barriers' by engaging stakeholders in the PHC systems planning, implementation, monitoring, and evaluation.

The role of empowering individuals, families and communities through community participation and engagement has been re-emphasised in connection with achieving the SDG target of UHC [152]. Our review shows that PHC in Ethiopia emphasises community empowerment and community participation as an important strategy to strengthen the PHC system and improve performance [5,126]. The national HEP has a significant component of community empowerment that is planned to be achieved by the HEWs through facilitating and participating in community service delivery [105]. In this review, community engagement was found to improve service uptake [112]. The review also shows that service user and caregiver involvement in mental health services improves the appropriateness and quality of services, promotes respect and ensures protection against mistreatment [153].

In our study, we observed that community participation improves the use of high-impact maternal and newborn health services, including skilled delivery care and postnatal care, with improved health system response [49]. Community participation also effectively promotes health behaviours, influences social norms, oversees health centers and provides support for community health workers [126]. In agreement with our study, a multi-country case study also shows that community empowerment through strengthening their participation to enhance access and equity of PHC services successfully [154]. Although there are many

informal ways of community empowerment, our study shows that the Women Development Army (WDA) [49], Health Development Army (HAD) and Pregnant Women Forum (PWF) [105] some of the social innovations were designed to empower service users and the community to participate in PHC service provision and increase service uptake. These micro-level organisations also have some level of community leadership activities with a significant potential to empower the community to contribute to the UHC in rural areas. A cross-country case study in LMICs shows that reinforcement of community empowerment through community co-learning, leadership, and accountability in the health system can be useful to achieve changes in the social and institutional system to support progress towards UHC [154].

Individual empowerment is significant in designing and providing healthcare at the PHC facilities. Empowered women are more likely to discuss and make decisions on the continuum of maternal care [155]. A recent scoping review also shows that people-centred PHC approaches – which could reflect individual and community empowerment, have been implemented in HICs. However, this study also shows little attention has been given to engaging end users to ensure people-centred PHC in LMICs [156]. Similarly, in Ethiopia, studies show that there are limited involvements and decision-making roles of schizophrenia clients in their care. Care providers were coercive in providing the required care, implying a lack of person-centred care [157]. There is a need to enforce evidence-based policy to improve individual and community empowerment and support the progress towards UHC. However, a PHC research priority-setting study in Uganda also shows that low community empowerment, weak governance and accountability for health promotion programmes are major challenges to set more policy-relevant evidence and actions to achieve UHC [158].

In this review, studies show that the national HEP is praised as the means to achieve most of the PHC goals, including community empowerment and engagement towards UHC [105] through facilitating community participation [12]. Nevertheless, the challenge remains in enhancing the capacity of health posts to meet increasing demand, improving the productivity and efficiency of health extension workers (HEWs), and involving the broader community to ensure effective community empowerment [12]. There is a need to equip service users, and service providers with the necessary skills and involve the local community in the process of care provision [153]. Given that community empowerment and active participation are critically important to achieving UHC, our study shows that supportive supervision and focused training can increase the capacity and motivation of the frontline HEWs to work more on community participation and empowerment [105]. There is limited understanding of the potential contribution of individual- or community-level empowerment [153,159]. Community involvement is rarely implemented with interventions to support individual or community engagement, particularly to strengthen rural health systems [159]. Studies also show that there is limited practical guidance available on how community empowerment can be achieved and sustained [154].

The strength of this scoping review is that it highlights the successes, challenges, and pathways towards UHC using the WHO's PHC Monitoring Framework. This framework contextualises findings within global health priorities and standards in a structured approach to evaluate and analyse the PHC implementation status in Ethiopia. Results from this study provide a comprehensive and structured understanding for the researcher, policymakers and program experts to identify, research, and resolve the challenges of achieving UHC and health security. However, we acknowledge that this study has some limitations. Ascertaining the strength of the evidence presented is challenging as quality assessment of the included studies is not compulsory for a scoping review. However, our robust data search strategy, screening, data quality assurance techniques and thematic analysis might have minimised this limitation. It is also possible that studies that might have been published in languages other than English

are excluded. While this study focused on national-level PHC system implementation, we might have included local-level data due to the paucity of nationally representative research findings. Therefore, due to the complexities of PHC in Ethiopia, our findings need contextual understanding and cautious conclusions and warrant further empirical research.

## Conclusions

Ethiopia's PHC system has made significant strides towards achieving UHC, particularly through the integration of key services such as family planning, HIV, and mental health into PHC. These achievements have been driven by strong governance and political commitment, notably through the National HEP, which has improved service access and strengthened community engagement. However, challenges persist, particularly in the insufficient prioritisation of NCDs, palliative care, and the uneven allocation of resources and infrastructure, especially in rural areas. Financial constraints, weak multisectoral collaboration, and limited monitoring and evaluation mechanisms continue to hinder progress. Targeted investments, governance reforms, and enhanced service integration are essential to overcome these obstacles. Strengthening multisectoral policies, improving partnerships, strengthening financial governance and prioritising NCD prevention and control programs in planning and resource allocation are crucial steps toward achieving UHC. Additionally, empowering communities and scaling up training for local leaders will ensure broader participation and support for PHC initiatives. Given that evidence-based decision-making in overcoming these challenges is a critical step in strengthening data collection and monitoring systems to guide policy adjustments is also important. Given that evidence-based decision-making in overcoming these challenges is a critical step strengthened data collection and monitoring systems to guide policy adjustments are also important. By addressing these systemic and infrastructural challenges with effective community engagement, multisectoral policy partnership and building on its successes, Ethiopia can further improve its PHC system, making it more resilient and better equipped to meet the health needs of its population, particularly in the face of emerging health threats.

## Supporting information

**S1 Table.  Search strategies.**
(DOCX)

**S2 Table.  Characteristics and key findings of the included studies.**
(DOCX)

**S3 Table.  The Arksey and O'Malley methodological framework for conducting a scoping study.**
(DOCX)

**S1 Checklist.  PRISMA checklist.**
(DOCX)

## Author contributions

**Conceptualization:** Yibeltal Assefa.

**Data curation:** Tesfaye Setegn Mengistu.

**Formal analysis:** Tesfaye Setegn Mengistu.

**Investigation:** Tesfaye Setegn Mengistu, Yibeltal Assefa.

**Methodology:** Tesfaye Setegn Mengistu, Yibeltal Assefa.

**Supervision:** Yibeltal Assefa.

**Validation:** Tesfaye Setegn Mengistu, Aklilu Endalamaw, Anteneh Zewdie, Eskinder Wolka, Yibeltal Assefa.

**Visualization:** Tesfaye Setegn Mengistu.

**Writing – original draft:** Tesfaye Setegn Mengistu.

**Writing – review & editing:** Aklilu Endalamaw, Anteneh Zewdie, Eskinder Wolka, Yibeltal Assefa.

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
