## [Decision Letter · Decision Letter 0]

8 Jan 2025

PGPH-D-24-02538

Strengthening Primary Health Care in Ethiopia: A Scoping Review of the Successes, Challenges, and Pathways Towards Universal Health Coverage and Health Security Using the WHO’s PHC Monitoring Framework

Dear Dr. Mengistu,

Thank you for submitting your manuscript to PLOS Global Public Health. After careful consideration, we feel that it has merit but does not fully meet PLOS Global Public Health’s publication criteria as it currently stands. Therefore, we invite you to submit a revised version of the manuscript that addresses the points raised during the review process.

We look forward to receiving your revised manuscript.

Kind regards,

Damen Haile Mariam, MD, MPH, PhD

Academic Editor

Journal Requirements:

Additional Editor Comments (if provided):

Reviewer 1 -

General comment:

- The material reviewed could be considered relevant for understanding the general situation of PHC system implementation in Ethiopia. The review has also tried to highlight crucial issues on moving forward with the UHC agenda in Ethiopia. Such as:

- Lack of a progress in monitoring system,

- The need for strengthening integrated service delivery,

- Challenges related to governance and political commitment at lower governance level

- The need to invest in the resilience of health systems.

- However, there is limited evidence obtained through this review with regards to studies conducted using important indictors suggested by WHO’s PHC monitoring framework, particularly concerning multisectoral action and empowerment of people and communities.

- In addition, the manuscript should include background that explains the country’s context, and brief introduction about PHC monitoring framework to explain its relevance to evaluate successes and challenges towards UHC and health security.

Methods:

- In this review, authors cannot claim they have strictly used the WHO’s PHC Monitoring Framework. The review could be taken as an assessment of the general situation related to Primary Health care implementation towards achieving UHC by reviewing available evidence and help in identifying information gaps. In addition, the classification of the studies retrieved should have been based on the indicators recommended in the WHO’s PHC Monitoring Framework. The classification made on the type of study does not indicate that the study of success ad challenges is based on the WHO framework specifically. Hence the need to modify the title.

- There are two most important indicators used to monitor progress towards UHC i.e., service coverage index (SCI) and financial risk protection. Important document I recommend is: Tracking Universal Health Coverage in the WHO African Region, 2022 WORLD HEALTH ORGANIZATION REGIONAL OFFICE FOR AFRICA BRAZZAVILLE — 2022. Also (Primary Health Care on the Road to Universal Health Coverage 2019 GLOBAL MONITORING REPORT). This review also excluded global studies and important documents that would be useful for comparison and for exchanging experiences between countries.

Results:

- In terms of studies retrieved, it would have been more relevant if characteristics of the studies have been classified based on their relevance to the important indicators provided by the WHO PHC Monitoring framework.

- In terms of governance, there is increasing focus on organizing service delivery at the district level. Accordingly, Ethiopia’s Woreda or District level transformation agenda has emphasized the need to create Primary Health Care Unite (PHCU) comprising five satellite HPs, referral HCs and primary hospital with activities planned and monitored at District /Woreda level. The authors do not seem to follow this definition of PHCU.

- In any case it is important to monitor progress and to find out the successes and challenges of implementing this extremely important initiative that would be necessary for strengthening continuum of care and the referral system. Hence, I would like to recommend inclusion of review of studies conducted in relation to district level management, such as that of AMREF: I.E (Woreda Level Deep Dive Assessment to Inform Integrated Health System Strengthening (IHSS) Investment, MERQ Consultancy PLC in Collaboration.

- In terms of specific needs of the population, most of the papers reviewed described important areas of PHC services but very few studies present findings of investigations that describe the actual situation in the country. It is noted that six studies described the importance of tailoring healthcare services and interventions to meet the specific health needs of a population. However, where is the evidence with regards to the actual situation in Ethiopia in this regard?

Discussion:

- The review was able to highlight primary strategic focus of the health system to achieve UHC. However, it lacked specific evidence to show the situation in the country due to shortage of focused studies based on WHO’s PHC monitoring framework and specifically to guide implementation of important initiatives.

- The paper has rightly emphasized the need for multisectoral approach to meet the dynamic population health needs. This study illustrates that multisectoral engagement and public-private partnerships within PHC systems remain fragmented and inconsistent. Only two papers looked into this issue, and they are not designed to evaluate the contribution of the approach. Presenting country’s specific experience such as the Health Sector Development Program (HSDP) a brief review of Ethiopia’s "Seqota Declaration" which could serve as exemplary strategy for muli-sectoral approach if implemented at scale.

- The importance of community participation is noted. From my own past experiences working in a program that supported strengthening of the HEP whose primary focus has been on creating access to primary health care services. It improved equity and quality of health care through utilization of essential health services, and building community ownership particularly in empowering women, as evidenced by achieving most of the health-related Millennium Development Goals (MDGs) has been great achievement of Ethiopia. But the sustainability of successes in this area needs properly structured monitoring and continued external reviews.

- The statement such as “The strength of this scoping review is that it provided a detailed landscape of PHC in Ethiopia by highlighting the successes, challenges, and pathways towards UHC and health security using the WHO’s PHC Monitoring Framework” cannot be accepted. Limitations in finding relevant studies needs to be emphasized.

- The statement … “Results from this study provide a comprehensive and structured understanding for the researcher, policymakers and program experts to identify, research, and resolve the challenges of achieving UHC and health security”. This is acceptable with the limitation of the review well explained.

- Clear recommendations are expected from the authors. There is a need to highlight areas that require in-depth studies: Such as the monitoring of the implementation of Woreda/District level transformation agenda, a very important initiative to strengthen Primary Health Care Unit establishment and functioning.

- As it could be understood from browsing the references listed in this paper there is lack of periodic assessment of the PHC system particularly to determine capacity of PHC for implementing UHC in Ethiopia. One of the most relevant studies cited is reference number 81 (Eregata GT, Hailu A, Memirie ST, Norheim OF. Measuring progress towards universal health coverage: National and subnational analysis in Ethiopia. BMJ Global Health 2019). In this paper it is stated that “… Few national and subnational studies monitor UHC in low-income countries and there is none for Ethiopia”.

- Concerning service integration, a need for political commitment and investment at all health system levels for Strengthening NCD prevention and control through PHC in Ethiopia requires emphasis as part of the recommendation.

Conclusion:

- It is noted that the objective of this review is to evaluate success and challenges towards achieving UHC by using WHO’s PHC Monitoring Framework. However, the finding is that studies designed to monitor the progress underway to ensure the implementation of relevant and attractive health polices is almost non-existent.

- However, the reviewers should be appreciated for have coming up with extensive review that describes what needs to be undertaken to implement a strong PHC services and also the need for monitoring system and stress that there is serious lack of focused studies that could help in evaluating success and challenges towards achieving UHC.

- There therefore there is a need to revise the presentation of this manuscript to show to Policy Makers and health partners the need for establishing a strong monitoring system by stressing the scarcity of data based on WHO’s PHC Monitoring Framework. Objective of the review should then be to evaluate availability of published studies to review the Successes, Challenges, and Pathways Towards Universal Health Coverage and Health Security Using the WHO’s PHC Monitoring.

- Ethiopia’s experience in regularly reviewing HSDP by using external review till the end should have been maintained. It should be remembered that it has been the review of HSDP I that initiated the Health Extension Program (HEP) and subsequent reviews helped in strengthening its implementation.

Reviewer 2-

General comment:

The synthesis of the literature offers a valuable foundation for understanding the landscape of primary health care in Ethiopia. However, I would like to raise a few points regarding the scope and depth of the analysis presented in your review. While the detailed summaries of the studies offer essential insights, it appears that the review primarily focuses on these narratives without incorporating a broader range of data elements like primary health care in hospital settings. This limitation could potentially hinder the overall depth of insight and analysis that can be derived from your findings. The inclusion of more extensive data would not only enrich the discussion but also foster a more nuanced understanding of the trends and patterns observed in the literature.

The time frame covered in your review—from 1998 to the present—provides a significant opportunity for reflection on the advancements and developments that have taken place within the primary health care system in Ethiopia over the past three decades. This longitudinal perspective is crucial, as it allows for the identification of both improvements and persistent challenges in the health care landscape.

To further enhance the impact and utility of your findings, I encourage you to incorporate additional data elements that could address the gaps and limitations identified in the reviewed studies. For instance, exploring the strengths of the primary health care system in Ethiopia, along with an examination of primary health care service delivery in hospital settings, could provide a more balanced view of the current state of health care delivery. By integrating these elements into your review, you would not only strengthen your analysis but also offer readers a more comprehensive understanding of the current landscape in your area of study. This would ultimately contribute to the ongoing dialogue surrounding primary health care in Ethiopia and help inform future efforts aimed at improving health outcomes.

Title:

- The title looks slightly overwhelming due to its length and complexity. The multiple concepts it introduces primary health care (PHC), successes, challenges, pathways, universal health coverage

(UHC), health security, and the WHO framework make it dense. Simplifying the title or breaking it into more focused elements could enhance readability and clarity, helping readers quickly grasp the core

focus of the manuscript. Additionally, the inclusion of both UHC and health security may be seen as broad, potentially requiring more specific.

- It would be beneficial to streamline the content by focusing on the central themes and eliminating redundancy. Breaking the title into more focused elements will help readers immediately grasp the main subject without being overwhelmed. Additionally, narrowing the scope to either "Universal Health Coverage" or "Health Security" would allow for a more precise focus, given that both are broad and may not be fully addressed in a single review. A more concise version could emphasize the key concepts of the review (Ethiopia's PHC) and its alignment with global health goals.

- Therefore, I suggest the title to be: "Strengthening Primary Health Care in Ethiopia: A Scoping Review of Successes, Challenges, and Pathways Towards Universal Health Coverage using WHO's Monitoring Framework".

Abstract:

- The abstract would benefit from more concrete suggestions for overcoming the identified challenges, such as strategies for improving resource allocation, addressing rural inequities, and fostering stronger governance and service integration.

- In addition, offering policy recommendations based on the findings could provide practical guidance for decision-makers in Ethiopia to effectively address the challenges and build on the successes of the PHC system.

Introduction:

- To enhance the introduction, it would be beneficial to incorporate more specific details about the ongoing challenges Ethiopia faces in its PHC system, especially in terms of regional disparities, resource allocation, and workforce distribution being impacted by narrow fiscal space. These aspects would provide a more balanced view of both the successes and challenges within the system. Additionally, a more in-depth discussion of the health extension program and its impact on improving health access to basic health services and outcomes in underserved areas would strengthen the narrative and provide concrete examples of Ethiopia's PHC approach in practice.

- Including case studies or specific instances of multisectoral collaboration, both successful and challenging, would offer readers a more understanding of how these partnerships work in practice and their impacts.

Methods:

- To improve the clarity and robustness of the methodology, expanding the inclusion criteria to include grey literature, such as government reports, policy documents, and publications from development partners or non-governmental organizations, could enrich the review by providing context-specific insights that are often omitted from peer-reviewed studies.

- To minimize potential language bias, it might be worth considering the inclusion of studies published in other languages, such as Amharic, or providing a justification for excluding non-English studies if this is not feasible.

- Additionally, involving more authors in the data extraction and verification process would enhance the reliability of the results and reduce the potential for bias or error. Finally, providing a clearer

description of the thematic analysis process, including how themes were identified and the specific subthemes used for categorization, would help.

Results:

- 1. Addressing Placeholder References: The manuscript should ensure that all references (e.g., Error! Reference source not found") citations to maintain the quality and professionalism of the manuscript.

- 2. Clarification and Elaboration on Key Findings: It would enhance the clarity and impact of the results to provide more detailed, concrete examples, especially in the discussion of challenges like the low

integration of NCD and palliative care services. Specific case studies or quantitative data would strengthen these claims and offer readers a clearer understanding of the situation.

- 3. Expansion on Digital Health: Given the increasing importance of digital technologies in healthcare, the section on digital health should be expanded to provide more in-depth analysis. Discussing

successful mHealth interventions, their impact on health outcomes, and the barriers to digital health adoption would make this section more robust. Addressing the role of digital health in improving

service delivery during the COVID-19 pandemic is particularly relevant and could be expanded.

- 4. Structuring Health Workforce Findings: The section on the health workforce could be more structured by categorizing the challenges into distinct themes such as staff shortages, turnover, burnout,

and training gaps, as well as successes in health workforce development and deployment. This would improve readability and allow for a more systematic discussion of the issue.

- 5. Strengthen Analysis on Health Security: There is a noted gap in the analysis of health security in the PHC system, which could be addressed more explicitly. Expanding on the relationship between

health security and PHC, including potential risks and the integration of health security measures within the PHC framework, would be valuable for readers concerned with health

system resilience.

- 6. Provide Clearer Data on Service Coverage and Inequality: The results section briefly touches on service coverage and inequality but lacks a more detailed breakdown. Adding figures or examples to

illustrate the extent of disparities in access to PHC services across different regions, urban vs. rural areas, and among different population groups would further underscore the gaps and provide

more actionable insights.

- 7. Improving Transitions between Sections: The review could benefit from smoother transitions between themes, ensuring that the findings related to each topic are logically connected and flow

coherently from one section to the next. This would improve readability and help the reader easily follow the narrative.

Discussion:

- To strengthen this discussion section, it is recommended to include more detailed examples of the political and governance challenges at various levels of the healthcare system, particularly focusing on

how regional and district-level leadership can be more effectively engaged in the implementation of PHC policies.

- Additionally, expanding the analysis of the barriers to service integration, particularly related to infrastructure and human resources, would provide a more holistic understanding of the challenges Ethiopia faces in expanding and sustaining PHC services. Given the importance of digital health and ICT in the future of PHC, a more detailed exploration of Ethiopia's ICT challenges -

such as internet access, data quality, and technology adoption would be useful.

- Finally, the manuscript could benefit from a more explicit connection between the identified challenges and potential solutions, such as recommending specific actions or policy adjustments that could address the governance issues, improve infrastructure, and enhance multisectoral coordination. Expanding the call for empirical research to include underexplored topics like the impact of service integration on PHC system resilience in emergencies and detailing how to engage the broader community in sustaining PHC, would help make the paper even more actionable for future research and policy development.

Conclusion:

- To enhance the conclusion section, it would be helpful to provide more specific, actionable recommendations for addressing the highlighted challenges. For instance, the manuscript could offer examples of successful governance reforms or multisectoral collaborations from other countries that could be adapted to the Ethiopian context. Concrete strategies for improving financial governance, such as

detailed steps to enhance funding allocation or innovative funding models for rural health services, could be included. The mention of NCDs, palliative care, and infrastructure gaps should be expanded upon with more concrete policy suggestions, such as prioritizing these areas in health planning and resource allocation.

- Additionally, the role of evidence-based decision-making in overcoming these challenges could be incorporated, perhaps suggesting the need for strengthened data collection and monitoring systems to guide policy adjustments.

- Lastly, while the empowerment of communities and local leaders is an important point, the manuscript could be improved by offering more details on how these efforts can be scaled, sustained and integrated into the broader PHC system, along with examples of effective community engagement models. By adding these specifics, the conclusion would offer a more roadmap for future progress in Ethiopia's PHC system.

Reviewers' comments:

Reviewer's Responses to Questions

**Comments to the Author**

1. Does this manuscript meet PLOS Global Public Health’s publication criteria ? Is the manuscript technically sound, and do the data support the conclusions? The manuscript must describe methodologically and ethically rigorous research with conclusions that are appropriately drawn based on the data presented.

Reviewer #1: Partly

Reviewer #2: Yes

2. Has the statistical analysis been performed appropriately and rigorously?

Reviewer #1: N/A

Reviewer #2: N/A

3. Have the authors made all data underlying the findings in their manuscript fully available (please refer to the Data Availability Statement at the start of the manuscript PDF file)?

Reviewer #1: Yes

Reviewer #2: No

4. Is the manuscript presented in an intelligible fashion and written in standard English?

Reviewer #1: Yes

Reviewer #2: Yes

5. Review Comments to the Author

Reviewer #1: General Comment:

The manuscript should include background that explain the country’s context. Brief introduction about PHC monitoring framework to explain its relevance to evaluate successes and challenges towards UHC and health security.

The material reviewed could be considered relevant for understanding the general situation of PHC system implementation in Ethiopia. The review has also tried to highlight crucial issues on moving forward with the UHC agenda in Ethiopia. Such as:

- Lack of a progress in monitoring system,

- The need for strengthening integrated service delivery,

- Challenges related to governance and political commitment at lower governance level

- The need to invest in the resilience of health systems.

However, there are limited evidences obtained through this review with regards to studies conducted using important indictors suggested by WHO’s PHC monitoring framework, particularly concerning multisectoral action and empowerment of people and communities,

Methods:

In this review authors cannot claim they have strictly used the WHO’s PHC Monitoring Framework. The review could be taken as an assessment of the general situation related to Primary Health care implementation towards achieving UHC by reviewing available evidences and help in identifying information gaps. And also the classification of the studies retrieved should have been based on the indicators recommended in the WHO’s PHC Monitoring Framework. The classification made on the type of study does not indicate that the study of success ad challenges is based on the WHO framework specifically. Hence the need to modify the title.

There are two most important indicators used to monitor progress towards UHC i.e service coverage index (SCI) and financial risk protection. Important document I recommend is: Tracking Universal Health Coverage in the WHO African Region, 2022 WORLD HEALTH ORGANIZATION REGIONAL OFFICE FOR AFRICA BRAZZAVILLE — 2022. Also (Primary Health Care on the Road to Universal Health Coverage 2019 GLOBAL MONITORING REPORT). But this review excluded global studies and these important documents are missed. Otherwise it should be realized that global studies are useful for comparison and could initiate exchange of experiences between countries.

Results:

Studies Retrieved:

Altogether, 110 studies were included in the review. Some were handpicked.

As suggested above it would have been more relevant if characteristics of the studies have been classified based on their relevance to the important indicators provided by the WHO PHC Monitoring framework.

Governance:

There is increasing focus on organizing service delivery at the district level. Accordingly, Ethiopia’s Woreda or District level transformation agenda has emphasized the need to create Primary Health Care Unite (PHCU) comprising five satellite HPs, referral HCs and primary hospital with activities planned and monitored at District /Woreda level. The study Team do not seem to follow this definition of PHCU.

In any case it is important to monitor progress and to find out the successes and challenges of implementing this extremely important initiative that would be necessary for strengthening continuum of care and the referral system. Hence I would like to recommend inclusion of review of studies conducted in relation to district level management, such as that of AMREF: I.E (Woreda Level Deep Dive Assessment to Inform Integrated Health System Strengthening (IHSS) Investment, MERQ Consultancy PLC in Collaboration

Specific needs of the population:

Most of the papers reviewed described important areas of PHC services but very few studies present findings of investigations that describe the actual situation in the country. It is noted that six studies described the importance of tailoring healthcare services and interventions to meet the specific health needs of a population. Yes, but where is the evidence with regards to the actual situation in Ethiopia in this regard?

Discussion:

The review was able to highlight primary strategic focus of the health system to achieve UHC. But lacked specific evidences to show the situation in the country due to shortage of focused studies based on WHO’s PHC monitoring framework and specifically to guide implementation of important initiatives.

- The paper has rightly emphasized the need for multisectoral approach to meet the dynamic population health needs. This study illustrates that multisectoral engagement and public-private partnerships within PHC systems remain fragmented and inconsistent. Only two papers looked into this issue and they are not designed to evaluate the contribution of the approach. Presenting country’s specific experience such as the Health Sector Development Program (HSDP) a brief review of Ethiopia’s Sequota Declaration which could serve as exemplary strategy for mulisecoral approach if implemented at scale.

- The importance of community participation is noted. From my own past experiences working in a program that supported strengthening of the HEP whose primary focus has been on creating access to primary health care services. It improved equity and quality of health care through utilization of essential health services, and building community ownership particularly in empowering women, as evidenced by achieving most of the health-related Millennium Development Goals (MDGs) has been great achievement of Ethiopia. But the sustainability of successes in this area needs properly structured monitoring and continued external reviews.

The statement such as “ The strength of this scoping review is that it provided a detailed landscape of PHC in Ethiopia by highlighting the successes, challenges, and pathways towards UHC and health security using the WHO’s PHC Monitoring Framework” cannot be accepted. Limitations in finding relevant studies needs to be emphasized.

- The statement … “Results from this study provide a comprehensive and structured understanding for the researcher, policymakers and program experts to identify, research, and resolve the challenges of achieving UHC and health security”. This is acceptable with the limitation of the review well explained.

- Clear recommendations are expected from the authors. There is a need to highlight areas that require in-depth studies: Such as the monitoring of the implementation of Woreda/District level transformation agenda, a very important initiative to strengthen Primary Health Care Unit establishment and functioning.

- As it could be understood from browsing the references listed in this paper there is lack of periodic assessment of the PHC system particularly to determine capacity of PHC for implementing UHC in Ethiopia. One of the most relevant study cited is reference number 81 (Eregata GT, Hailu A, Memirie ST, Norheim OF. Measuring progress towards universal health coverage: National and subnational analysis in Ethiopia. BMJ Global Health 2019). In this paper it is stated that “… Few national and subnational studies monitor UHC in low-income countries and there is none for Ethiopia”.

- Concerning service integration, a need for political commitment and investment at all health system levels for Strengthening NCD prevention and control through PHC in Ethiopia requires emphasis as part of the recommendation.

Conclusion:

It is noted that the objective of this review is to evaluate success and challenges towards achieving UHC by using WHO’s PHC Monitoring Framework. However, the finding is that studies designed to monitor the progress underway to ensure the implementation of relevant and attractive health polices is almost non-existent.

However, the reviewers should be appreciated for have coming up with extensive review that describes what needs to be undertaken to implement a strong PHC services and also the need for monitoring system and stress that there is serious lack of focused studies that could help in evaluating success and challenges towards achieving UHC.

There therefore there is a need to revise the presentation of this manuscript to show to Policy Makers and health partners the need for establishing a strong monitoring system by stressing the scarcity of data based on WHO’s PHC Monitoring Framework. Objective of the review should then be to evaluate availability of published studies to review the Successes, Challenges, and Pathways Towards Universal Health Coverage and Health Security Using the WHO’s PHC Monitoring.

Ethiopia’s experience in regularly reviewing HSDP by using external review till the end should have been maintained. It should be remembered that it has been the review of HSDP I that initiated the Health Extension Program (HEP) and subsequent reviews helped in strengthening its implementation.

Reviewer #2: I would like to express my sincere appreciation for the comprehensive summary you have provided regarding the 110 studies included in your scoping review. Your synthesis of the literature offers a valuable foundation for understanding the landscape of primary health care in Ethiopia. However, I would like to raise a few points regarding the scope and depth of the analysis presented in your review.

While the detailed summaries of the studies offer essential insights, it appears that the review primarily focuses on these narratives without incorporating a broader range of data elements like primary health care in hospital settings. This limitation could potentially hinder the overall depth of insight and analysis that can be derived from your findings. The inclusion of more extensive data would not only enrich the discussion but also foster a more nuanced understanding of the trends and patterns observed in the literature.

The time frame covered in your review—from 1998 to the present—provides a significant opportunity for reflection on the advancements and developments that have taken place within the primary health care system in Ethiopia over the past three decades. This longitudinal perspective is crucial, as it allows for the identification of both improvements and persistent challenges in the health care landscape.

To further enhance the impact and utility of your findings, I encourage you to incorporate additional data elements that could address the gaps and limitations identified in the reviewed studies. For instance, exploring the strengths of the primary health care system in Ethiopia, along with an examination of primary health care service delivery in hospital settings, could provide a more balanced view of the current state of health care delivery.

By integrating these elements into your review, you would not only strengthen your analysis but also offer readers a more comprehensive understanding of the current landscape in your area of study. This would ultimately contribute to the ongoing dialogue surrounding primary health care in Ethiopia and help inform future efforts aimed at improving health outcomes.

6. PLOS authors have the option to publish the peer review history of their article (what does this mean? ). If published, this will include your full peer review and any attached files.

**Do you want your identity to be public for this peer review?** For information about this choice, including consent withdrawal, please see our Privacy Policy .

Reviewer #1: No

Reviewer #2: **Yes: ** Kebede Worku

---

## [Editor Report · Decision Letter 1]

14 Mar 2025

Strengthening Primary Health Care in Ethiopia: A Scoping Review of Successes, Challenges, and Pathways Towards Universal Health Coverage Using the WHO Monitoring Framework

PGPH-D-24-02538R1

Dear Dr. Mengistu,

We are pleased to inform you that your manuscript 'Strengthening Primary Health Care in Ethiopia: A Scoping Review of Successes, Challenges, and Pathways Towards Universal Health Coverage Using the WHO Monitoring Framework' has been provisionally accepted for publication in PLOS Global Public Health.

Best regards,

Damen Haile Mariam, MD, MPH, PhD

Academic Editor